# Missed and Detected Incidental Breast Cancers on Contrast Enhanced Chest CT: Detection Rates and CT Features

**DOI:** 10.3390/diagnostics13091522

**Published:** 2023-04-23

**Authors:** Jiyeon Park, Chohee Kim, Yoon Ki Cha, Myung Jin Chung

**Affiliations:** 1Department of Radiology and Center for Imaging Science, Samsung Medical Center, Sungkyunkwan University School of Medicine, Seoul 06351, Republic of Korea; jiyeonp33@gmail.com (J.P.);; 2Department of Radiology, Ansan Hospital, Korea University College of Medicine, Seoul 02841, Republic of Korea

**Keywords:** multidetector computed tomography, breast neoplasms, thorax

## Abstract

This study investigated the rate at which radiologists miss or detect incidental breast cancers on chest CT and to compare the CT features between the two groups. This retrospective study evaluated chest CT examinations and medical records of patients who registered with the diagnosis code of “breast cancer” between January 2016 and December 2020, and who had undergone contrast enhanced chest CT 3–18 months before registration, during which they were unaware of any breast lesions. This study found that out of 84 patients, incidental breast cancer lesions were missed in 54 (64.3%) and detected in 30 (53.7%). The initial treatment was delayed in the missed breast lesions group (*p* = 0.004). Breast lesions of smaller sizes (<9.0 mm, *p* = 0.01), or with lower enhancement ratios (<1.4, *p* = 0.009), were more likely to be missed. When three radiologists re-read the CTs with more attention to breast area, they detected breast cancers with higher accuracies (90.1%, 87.9%, and 81.3%). In summary, this study revealed that radiologists miss 64.3% of incidental breast cancers on chest CT, especially those of sub-centimeter sizes and weak enhancements.

## 1. Introduction

Breast cancer is the leading cause of cancer-related death among women, and its early detection is critical for improving outcomes [1,2]. Mammography, sonography, and MRI are the standard imaging modalities used for diagnosing breast cancer. However, while chest CT captures the entire breast in the images, it is not a primary diagnostic examination for breast cancer. The sensitivity and specificity of chest CT for detecting breast cancer are lower compared to MRI and the risk of radiation exposure is also a concern [3,4]. Consequently, radiologists interpreting chest CT scans may not pay significant attention to the breast region.

Despite this, the incidence of incidental breast lesions on chest CT is increasing, as the number of chest CT scans performed annually continues to rise [5]. Previous studies have reported that incidental breast lesions are found in 0.7% to 7.63% of chest CT scans [6,7,8] and between 31.0% to 69.6% of these lesions are malignant upon pathological examination [4,9]. In particular, breast lesions with irregular margins and high enhancement on chest CT are more likely to be suggestive of malignancy [5,10].

This study aims to investigate the rates at which incidental breast cancers are missed or detected on chest CT scans by radiologists. By analyzing the CT features of the missed and detected lesions, this study aims to improve the accuracy and timeliness of breast cancer diagnosis, ultimately leading to better patient outcomes. It may also identify specific imaging characteristics that could be used to develop future artificial intelligence systems for detecting breast cancer on chest CT scans.

## 2. Materials and Methods

### 2.1. Compliance with Ethical Standards

The Institutional Review Board approved this retrospective study (IRB # 2021-08-082-001). The requirement for written informed consent from the patients was waived.

### 2.2. Patients

A retrospective review of patients with breast cancer was performed at a single tertiary hospital. Eligibility criteria were: (1) patients registered with the diagnosis code of “breast cancer” in the electronic medical record of our hospital between January 2016 and December 2020; (2) patients who had undergone contrast enhanced chest CT 3–18 months before the breast cancer diagnosis; and (3) breast cancer lesions existed on the CT and the patients were unaware of having breast cancer at the time of the CT examination.

The reasons for the patients’ chest CT examinations were investigated in the medical record. All patients underwent breast biopsy or surgery, and breast cancer was pathologically diagnosed. The initial treatment method was investigated, and the time delay between the chest CT examinations and the initial treatment was calculated for each patient.

### 2.3. Chest CT Scan Protocol

Chest CT was performed at our institution using various multidetector CT scanners after contrast material injection. Scanning was performed using the helical technique with a 1.25-, 2.5-, or 3-mm reconstruction interval. IV contrast medium injection was given in all patients: 1.5 mL/kg of body weight was injected. The interval time between the initiation of contrast injection and scanning was 45 s.

### 2.4. Assessment of Breast Cancers on Chest CT and Other Breast Imaging Modalities

One reviewer, with two years’ experience in chest radiology, reviewed the chest CTs of the study patients to determine whether breast cancer lesions were present at the time of the examination. An enhancing breast lesion was defined as a breast lesion with an attenuation higher than that of the normal breast glandular tissue. On the chest CT, the reviewer looked for enhancing breast lesions and recorded the location (clockwise direction and distance from the nipple), number of the lesions, and size (in mm). Then the reviewer compared the chest CT with other breast imaging examinations of the patient, including breast MRI and sonography, and the surgical or pathological report. If the imaging features of the enhancing breast lesion on chest CT were identical to those on other breast imaging modalities and surgical or pathological reports, the patient was considered to have a breast cancer lesion at the time of the chest CT.

The patients were divided into two groups according to the initial report of the chest CT: missed and detected breast cancer lesions.

### 2.5. Analysis of Imaging Findings of Breast Cancer on Chest CT

The following CT findings were investigated for each lesion: number of enhancing lesions, size, enhancement ratio, breast density, shape (round, oval, or irregular), margin (well-circumscribed or not well-circumscribed), and enhancement pattern (homogeneous or heterogeneous) [11]. The longest diameter of the largest lesion was manually measured. For the breast lesion enhancement, the enhancement ratio of the breast lesion, defined as the ratio of Hounsfield units of the enhancing portion of the breast lesion to that of the ipsilateral latissimus dorsi or serratus anterior muscle, was calculated [10]. To evaluate breast density, Breast Imaging Reporting and Data System (BI-RADS) grades for mammography were applied because breast density readings on chest CT were consistent with those on mammography [9]. Breast density was classified into four grades.

### 2.6. Re-Assessment of the Detection Rate of Breast Cancers on Chest CT

Three specialized radiologists (with 30-, 15-, and 9-years’ experience in chest CT interpretation, respectively) re-read the group of chest CT examinations of both our study patients and the negative controls. The negative controls included chest CTs of seven breast cancer patients who did not have enhancing breast lesions on the chest CTs. During the re-reading process, the radiologists were instructed to concentrate on the breast area and to report any breast lesions that appeared suspicious for breast cancer. The accuracy of detecting breast cancers on CT scans was assessed for each reviewer.

### 2.7. Statistical Analysis

The demographic and clinical characteristics of the groups were compared using appropriate statistical tests, such as the Student’s *t*-test, Wilcoxon rank-sum test, chi-square test, or Fisher exact test. Categorical variables, such as breast density, shape, margin, and enhancement pattern, were expressed as frequencies in percentages. The cut-off values for the size and enhancement ratio were determined using ROC curve analysis. Multivariable analyses were conducted to identify significant factors associated with missed breast lesions. The statistical software SAS version 9.4 (SAS Institute, Cary, NC, USA) was used for all analyses. The significance level for all analyses was set at *p* < 0.05.

## 3. Results

We identified 342 patients who had registered with the diagnosis code of “breast cancer” to the medical records of our hospital between January 2016 and December 2020 and who had undergone chest CT examinations 3–18 months before the date of disease code registration. Among them, 251 patients were excluded as the patients were aware of having breast cancer at the time of the CT examination, and an additional seven patients were excluded since there was no breast lesion on chest CT. Finally, 84 patients were enrolled (Figure 1).

Among the 84 patients, radiologists had missed breast lesions during the initial reading of 54 patients (64.3%). These patients were categorized into the missed breast lesions group (Figure 2). Conversely, radiologists had detected breast lesions during the initial reading of 30 patients (35.7%). These patients were categorized into the detected breast lesions group.

### 3.1. Characteristics of Missed Breast Lesions Group and Detected Breast Lesions Group

All 84 patients were women. The demographic and clinical features of the patients are summarized in Table 1. The mean ages of the patients were 58.2 ± 11.5 years and 61.8 ± 11.5 years, each in the missed and detected lesions groups, respectively. The breast cancer type and cancer stage did not show significant difference between the two groups. The median delays between chest CT examination and initial treatment were 6.7 months in the missed breast lesions group and 2.3 months in the detected breast lesions group. The delay in the missed breast lesions group was significantly longer than that of the detected breast lesions groups (*p* = 0.004). The treatment methods did not show a significant difference between the two groups. Sixty-five patients had surgery as their initial treatment. Comparatively, nine patients underwent chemotherapy since they had distant metastasis or another primary cancer. The remaining 10 patients did not undergo treatment due to old age, poor general condition, or were lost during follow up.

### 3.2. Imaging Features of Missed and Detected Breast Cancers

CT features of the missed and detected breast lesions were compared (Table 2, Figure 3 and Figure 4). The mean size of the missed breast lesions was 8.77 ± 4.01 (range, 2.2–21.7) mm, while that of the detected breast lesions was 10.75 ± 4.63 (range, 4.7–21.6) mm. The mean size of the missed breast lesions was significantly smaller than that of the detected breast lesions (*p* = 0.046). The mean enhancement ratio of the missed breast lesions was 1.37 ± 0.48 (range, 0.5–2.8), while that of the detected breast lesions was 1.55 ± 0.36 (range, 1.0–2.3). The detected breast lesions group showed a higher enhancement ratio than the missed breast lesions group (*p* = 0.02). Analyses of imaging features including number of lesions, location, shape, margin, and enhancement pattern are summarized in Table 2. These features did not show significant differences between the two groups.

When both groups were considered together, 30 patients (30/84, 35.7%) had round breast lesions; seven patients had oval breast lesions (7/84, 8.3%); and 47 patients (57/84, 56.0%) had irregularly shaped breast lesions. In addition, 42 patients (42/84, 50.0%) had breast lesions with well-circumscribed margins, whereas 42 patients (42/84, 50.0%) had lesions without well-circumscribed margins.

### 3.3. Imaging Features Associated with Missed Breast Lesions Group

The size and enhancement ratio showed significant differences between the two groups of missed and detected breast lesions group. The ROC curve was used to determine the cut-off value for missed breast lesions. Breast lesions of sizes greater than or equal to 9.0 mm were found in 60.0% of the detected lesions group and in 33.3% of the missed breast lesions group (*p* = 0.02). Breast lesions of enhancement ratio higher than or equal to 1.4 were found in 63.3% of the detected lesions group and in 37.0% of the missed breast lesions group (*p* = 0.02).

The multivariable logistic regression analysis revealed that the breast lesion size (<9.0 mm) and enhancement ratio (<1.4) were significant features associated with missed breast lesions (*p* = 0.01 and *p* = 0.009, each) (Table 3). Other features did not show significant associations.

### 3.4. Re-Assessment of the Detection Rate of Breast Cancers on Chest CT

Three radiologists specialized in chest radiology re-read the chest CT scans of 91 patients, of whom 84 were study patients and seven were negative controls. The first reviewer reported 77 positive lesions (true-positive). This reviewer missed breast lesions in six patients (false-negative) and reported incorrect lesions in three patients (false-positive). Their accuracy was 90.1%. The second reviewer reported 73 positive lesions (true-positive). This reviewer missed breast lesions in eight patients (false-negative) and checked for wrong lesions in three patients (false-positive). Their accuracy was 87.9%. The third reviewer reported 67 positive lesions (true-positive). This reviewer missed breast lesions in 17 patients (false-negative). Their accuracy was 81.3%.

## 4. Discussion

This retrospective study revealed that radiologists reading contrast enhanced chest CT missed 64.3% of incidental breast cancers, and that the breast cancers were later detected on other breast imaging modalities. Breast lesions < 9.0 mm or without intense enhancement (enhancement ratio < 1.4) were more likely to be missed on chest CT.

The incidence of incidental breast lesions on chest CT scans has been discussed. The detection rates vary across studies. Hussain et al. [8] reported 33 (7.6%) cases of incidental breast cancer detection among 431 chest CT scans. Lin et al. [5] studied numerous cohorts, involving 2251 chest CT scans, and reported only 16 (0.7%) cases of incidental breast cancer detection. Pathologic analyses of incidental breast lesions show a prevalence of malignancy of 31.0–69.6% [4,5].

However, to date, the rate at which radiologists miss incidental breast cancers on chest CT has not been investigated. In this study, we retrospectively reviewed chest CT performed before the cancer diagnosis in patients with breast cancer and found 84 cases of breast cancers on chest CT. Among these, only 35.7% were incidentally detected by radiologists in the initial report, whereas the other 64.3% were missed. In this study, when three radiologists re-read the CT scans knowing that the patients were later diagnosed with breast cancer, the detection rate increased significantly (accuracy of 90.1%, 87.9%, and 81.3%). Our study also revealed that the median delay between the CT examination and treatment initiation was significantly longer in the missed breast lesions group (6.7 months) than in the detected breast lesions group (2.3 months). These results suggest that when radiologists examined the breast area more carefully during the initial reading, they could have detected more incidental breast cancers, and have hastened the time of cancer diagnosis.

Furthermore, to the best of our knowledge, no study has compared CT features of missed or detected breast cancers using chest CT. ROC curve and multivariable analysis revealed that incidental breast lesions with a size of less than 9.0 mm or with an enhancement ratio lower than 1.4 were more likely to be missed on chest CT. The shape, margin, or enhancement pattern did not significantly affect the detection rate. These results suggest that radiologists should suspect the possibility of breast cancer even with small and poorly enhanced lesions.

Reliable CT features for incidental malignant breast cancer have been investigated. Choi et al. [10] reported that non-circumscribed margins and high contrast enhancement are favorable imaging features indicating malignancy of incidental breast lesions. Inoue et al. [12] reported that irregular margins and shapes are highly predictive features of malignancy in both mass and non-mass breast lesions. These results are consistent with those of a few studies analyzing breast MRI and suggest that breast cancers are more likely to have irregular shapes and spiculated margins [13]. However, Bach et al. [14] reported that incidental benign and malignant breast lesions are not safely distinguished from each other on chest CT. Although our study only included breast cancer patients, we revealed that 44.0% of breast cancers were round or oval, and approximately 50% of them had well-circumscribed margins. These results suggest that the shape and margin are insufficient predictive factors for malignancy on chest CT. We hypothesized that because chest CT has a larger field of view and provides less detailed visualization of soft tissue compared to breast MRI, it is not as precise as breast MRI in analyzing soft tissue lesions in the breast.

This study had a few limitations. First, this was performed retrospectively with a limited sample size. Since we only included patients who were unaware of breast cancers at the time of chest CT examinations, despite the larger cohort of subjects with breast cancers on chest CT, the number of exclusions was exceptional. Second, our study did not include enough negative controls. The radiologists who participated in the re-assessment knew that a large percentage of the patients involved in this study had breast cancer. This preconceived notion would have led to overdetection of small lesions that would have otherwise been ignored in a routine reading environment. As a result, the re-assessed detection rate of breast cancer may be higher. In addition, since our study only included positive controls when analyzing CT features of the breast lesions, we could not suggest specific methods to differentiate benign breast lesions from breast cancer lesions on chest CT. A further retrospective study is warranted to evaluate the difference.

## 5. Conclusions

This retrospective study demonstrated a 64.3% rate of missed incidental breast cancers on chest CT scans. This study also highlighted the importance of careful examination of the breast area during initial reading, as many small-sized or weakly enhancing breast lesions later turned out to be breast cancer. Our findings suggest that radiologists should be aware of the potential for incidental breast cancer detection on chest CT and recommend further evaluation through other imaging modalities. This study also identified specific CT features associated with missed incidental breast cancers, which may aid in the development of artificial intelligence systems capable of accurately detecting breast cancer on chest CT scans. These findings have significant clinical implications and may ultimately improve the accuracy and timeliness of breast cancer diagnosis, resulting in better outcomes for patients.

## Figures and Tables

**Figure 1 diagnostics-13-01522-f001:**
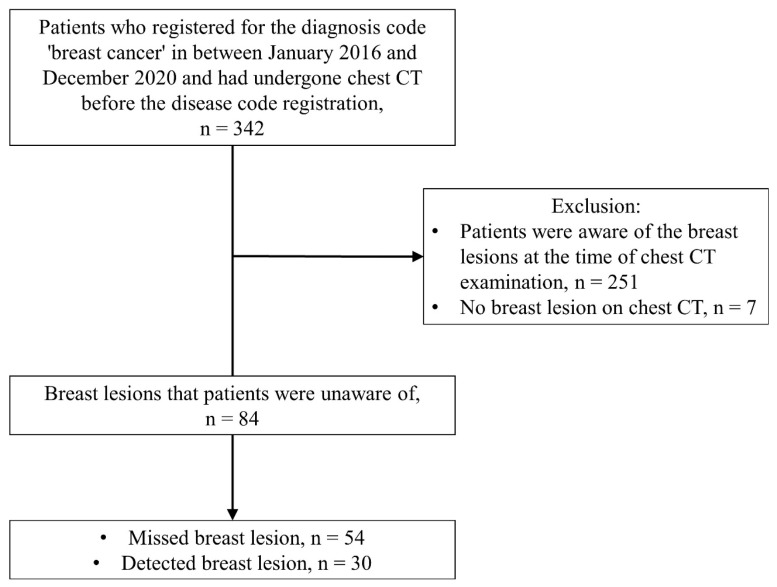
Flow diagram showing the patient selection process.

**Figure 2 diagnostics-13-01522-f002:**
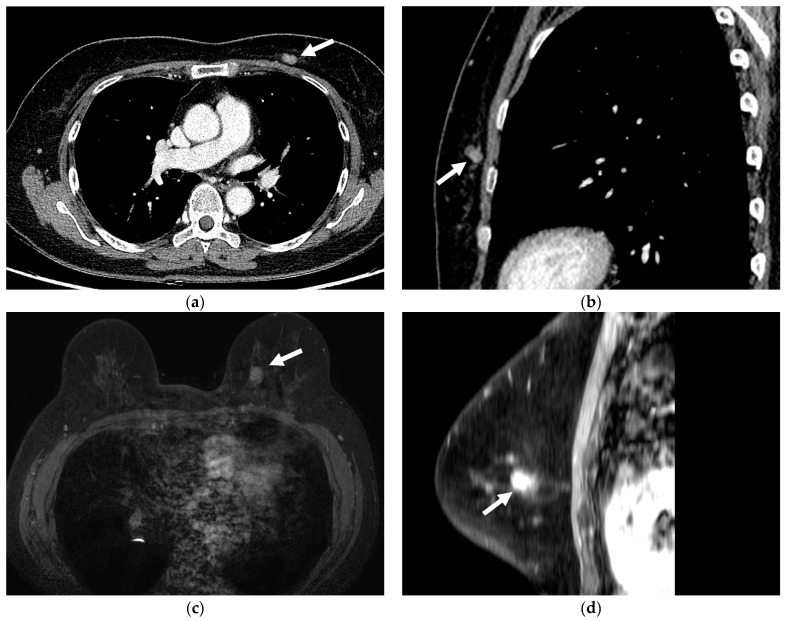
Contrast enhanced chest CT and breast MRI obtained in a 71-year-old female patient. (**a**) Axial, and (**b**) sagittal view of contrast enhanced chest CT obtained for metastasis workup of ovarian cancer show an 11.8-mm-sized enhancing mass at the 5 o’clock left breast (arrow). This mass was missed on the initial chest CT report. (**c**) Axial, and (**d**) sagittal view of contrast-enhanced breast MRI taken five months after the chest CT show an enhancing mass in the same position as that on chest CT (arrow). The patient was referred for a breast lesion detected on screening mammography. Mastectomy confirmed breast cancer.

**Figure 3 diagnostics-13-01522-f003:**
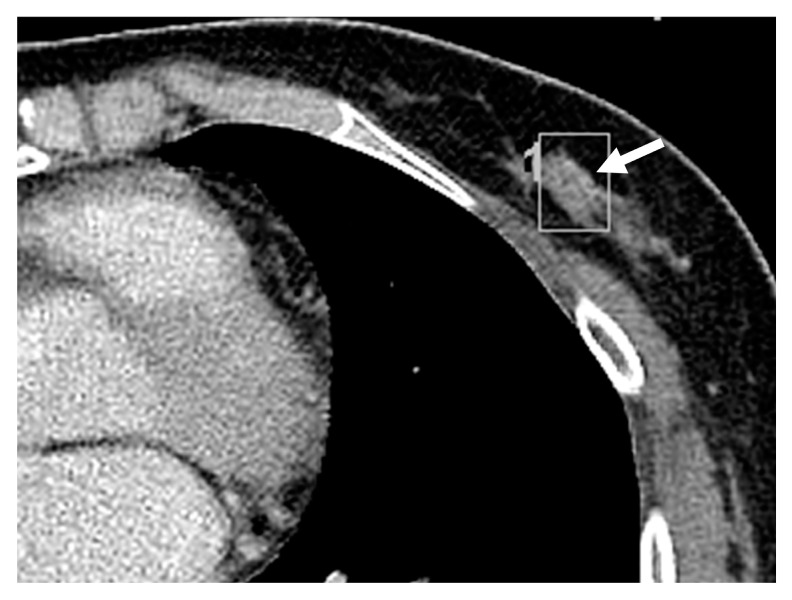
Contrast enhanced chest CT obtained of a 60-year-old woman whose incidental breast cancer was missed on the chest CT. Axial view of the chest CT obtained for urothelial cancer follow-up shows an 8.6-mm-sized, mildly enhancing, single mass at the left 6 o’clock breast (arrow). The mass is irregularly shaped, and the margin is not well-circumscribed. The mass is homogeneously enhanced with an enhancement ratio of 1.09. The breast density is BI-RADS grade 2. The mass was missed on the initial report. Twelve months later, the mass was detected on screening breast sonography.

**Figure 4 diagnostics-13-01522-f004:**
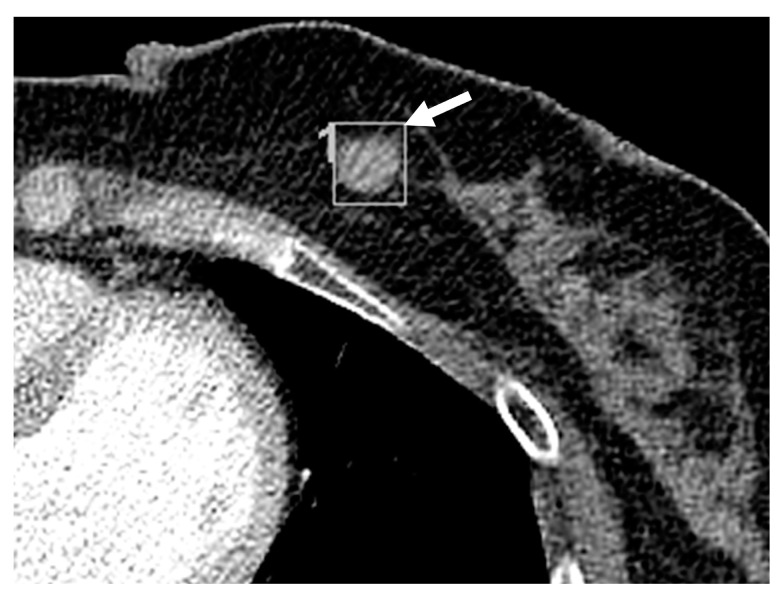
Contrast enhanced chest CT obtained of a 62-year-old woman whose incidental breast cancer was detected on the chest CT. Axial view of the chest CT obtained for lung cancer follow-up shows a 12.4-mm-sized single mass at the left 7 o’clock breast (arrow). The mass is round, and the margin is well-circumscribed. The mass is homogeneously enhanced with an enhancement ratio of 1.63. The breast density is BI-RADS grade 3. The mass was detected on the initial report.

**Table 1 diagnostics-13-01522-t001:** Demographic and clinical characteristics of the study patients.

Variable	Missed Breast Lesions (n = 54)	Detected Breast Lesions (n = 30)	*p*-Value
General characteristic			
Sex			
Female	54 (64.3)	30 (35.7)	
Mean age (years)	58.2 ± 11.5	61.8 ± 11.5	0.17
Reason for chest CT examinations			0.36
Malignancy	44 (52.3)	27 (32.1)	
Lung cancer	12 (14.3)	21 (25.0)	
Malignancy other than lung cancer	32 (38.1)	6 (7.1)	
Others	10 (11.9)	3 (3.6)	
Follow-up of lung nodules	3 (3.6)	1 (1.2)	
Pulmonary infection	3 (3.6)	0 (0)	
Chest pain or palpitation	2 (2.4)	1 (1.2)	
Others	2 (2.4)	1 (1.2)	
Breast cancer type			0.06
DCIS	13 (15.5)	2 (2.4)	
IDC	36 (42.9)	27 (32.1)	
Others	5 (6.0)	1 (1.2)	
Breast cancer stage			0.11
Stage 0	12 (14.2)	2 (2.4)	
Stage 1	25 (29.8))	15 (17.9)	
Stage 2,3,4	9 (10.7)	9 (10.7)	
Missing date	8 (9.5)	4 (4.8)	
Delay between chest CT scan and treatment			0.004
Mean ± SD (months)	9.08 ± 6.52	5.90 ± 6.58	
Median (months)	6.70	2.30	
Initial treatment			
Surgery	39 (46.4)	26 (31.0)	
Chemotherapy	7 (8.3)	2 (2.4)	
Follow up loss/No treatment	8 (9.5)	2 (2.4)	

Unless otherwise indicated, data are numbered with the percentage in parentheses. SD: standard deviation.

**Table 2 diagnostics-13-01522-t002:** CT features of study patients’ groups.

Variable	Missed Breast Lesions (n = 54)	Detected Breast Lesions (n = 30)	*p*-Value
Number of lesions			0.99
One	43 (79.6)	22 (60.6)	
Two	1 (1.9)	1 (3.3)	
Three or more	10 (18.5)	7 (23.3)	
Size			0.046
Mean ± SD (mm)	8.77 ± 4.01	10.75 ± 4.63	
Median (mm)	7.95	10.30	
Location			0.58
Right breast	24 (28.6)	16 (19.0)	
Left breast	30 (35.7)	14 (16.7)	
Shape			0.94
Round	20 (23.8)	10 (11.9)	
Oval	5 (6.0)	2 (2.4)	
Irregular	29 (34.5)	18 (21.4)	
Margin			0.50
Well-circumscribed	25 (29.8)	17 (20.2)	
Not well-circumscribed	29 (34.5)	13 (15.4)	
Enhancement pattern			0.27
Homogeneous	26 (31.0)	19 (22.6)	
Heterogeneous	28 (33.3)	11 (13.1)	
Enhancement ratio *			0.024
Mean ± SD	1.37 ± 0.48	1.55 ± 0.36	
Median	1.21	1.54	
Breast density			0.90
BI-RADS grade 1	6 (7.1)	5 (6.0)	
BI-RADS grade 2	32(38.1)	16 (19.0)	
BI-RADS grade 3	12 (14.3)	7 (8.3)	
BI-RADS grade 4	4 (4.8)	2 (2.4)	

Data presented as mean ± standard deviation, mean, or n (%). * Enhancement ratio: Hounsfield unit of the breast lesion/Hounsfield unit of the trapezius or serratus anterior muscle. BI-RADS = Breast Imaging Reporting and Data System, SD = standard deviation. Demographic and clinical characteristics of the study patients.

**Table 3 diagnostics-13-01522-t003:** Multivariable analysis of imaging features associated with missed breast lesions.

Variable	Odds Ratio (95% CI)	*p*-Value
Size [ref: ≥9.0 mm]	4.33 (1.40–13.35)	0.011
Location		
Right [ref: left]	0.74 (0.27–2.05)	0.56
Shape		
Oval [ref: Irregular]	5.07 (0.49–52.93)	0.18
Round [ref: Irregular]	2.49 (0.67–9.27)	0.18
Margin		
Well-circumscribed [ref: Not well-circumscribed]	1.03 (0.29–3.64)	0.96
Enhance pattern		
Homogeneous [ref: Heterogeneous]	0.4 (0.1–1.61)	0.20
Enhancement ratio * [ref: ≥1.4]	4.58 (1.46–14.4)	0.009
Breast density		
BI-RADS grade 2 [ref: Grade 1]	1.64 (0.36–7.59)	0.53
BI-RADS grade 3 [ref: Grade 1]	3.09 (0.49–19.42)	0.23
BI-RADS grade 4 [ref: Grade 1]	1.13 (0.11–11.74)	0.92

Odds ratios and *p*-values were obtained using logistic regression method. The reference category for each categorical variable is indicated by square brackets in the first column. * Enhancement ratio: Hounsfield unit of the breast lesion/Hounsfield unit of the trapezius or serratus anterior muscle. BI-RADS = Breast Imaging Reporting and Data System, CI = confidence interval.

## Data Availability

The data presented in this study are available on request from the corresponding author. The patients’ personal information, such as the patient number and name, have been deleted and may be disclosed if necessary.

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
