# Peer review of "Missed and Detected Incidental Breast Cancers on Contrast Enhanced Chest CT: Detection Rates and CT Features"

_diagnostics, 2023, doi:10.3390/diagnostics13091522_

Round 1

Reviewer 1 Report

The paper describes the retrospective steady of chest CT images of 84 patients with diagnosed breast cancer. The incidental breast cancer lesions were missed in 54 (64.3%) and detected in 30 (53.7%) patients. The comparison of the CT features between the two groups revealed that breast lesions with smaller sizes (<9.0mm, p=0.01) or with lower enhancement ratios (<1.4, p = 0.009) were more likely to be missed. Three experienced radiologists re-evaluated the CTs focusing on breast area and detected breast cancers with higher accuracies (90.1%, 87.9%, and 81.3%). The paper is well writeen and easy to follow. The figures and tables are clear, methods are adequately described. The limitations of the study are discussed.

I suggest a revision because of three reasons:

The main contributions of the research are to be summarized at the end of the introduciton section.

The p-values are provided in Tab.1-3 and several criteria are mentioned in Methods. I suggest to explicitly state the criterion for each p-value and mark significant case in some way in the tables.

The Conclusions section is to be added.

Also there are some minor issues:

lines 118-133 are to be removed

line 154 "This is a figure. Schemes follow the same formatting" is probably to  be removed

lines 246-248 "This section may be divided by subheadings. It should provide a concise and precise description of the experimental results, their interpretation, as well as the experimental conclusions that can be drawn." - are to be removed

Author Response

Thank you for reviewing our study. Below are the revisions we made based on your feedback.

REVIEWER 1

POINT 1 : The main contributions of the research are to be summarized at the end of the introduciton section.

RESPONSE 1 : We have summarized and clarified the purpose and contribution of our study in the final part of the introduction to better convey them. We also have revised the introduction to provide a clearer explanation of the role and challenges of chest CT in breast cancer diagnosis.

POINT 2 : The p-values are provided in Tab.1-3 and several criteria are mentioned in Methods. I suggest to explicitly state the criterion for each p-value and mark significant case in some way in the tables.

RESPONSE 2 : We have specified in our analysis that we used a significance level of p < 0.05, and have revised each table to highlight significant p values.

POINT 3 : The Conclusions section is to be added.

RESPONSE 3 : We have separated the conclusion from the discussion section as a distinct part. We also revised the conclusion to better convey the results of the study.

POINT 4 : Also there are some minor issues:

RESPONSE 4 : We have removed the unnecessary sentences. 

Reviewer 2 Report

This manuscript aims to investigate the rate at which radiologists miss or detect incidental breast cancers on chest CT and to compare the CT features between the two groups. The paper is quite of interest to readers, because the incidence of breast cancer is an important issue in women health, but it needs some improvements before being considered for publication.

In particular, the Introduction section must be totally re-written as it lacks of any scientific soundness!

Secondly, tables and figures should have a better layout. 

As a final consideration, Conclusion section should be separate from the rest of the paper. 

Author Response

Thank you for reviewing our study. Below are the revisions we made based on your feedback.

POINT 1 : In particular, the Introduction section must be totally re-written as it lacks of any scientific soundness!

RESPONSE 1: We have revised the introduction to explain the role and limitations of chest CT scans in breast cancer diagnosis, and highlighted the importance of our study in emphasizing the significance of chest CT scans. We have also explained that our study aimed to investigate the characteristics of breast cancer that can be identified through chest CT scans.

POINT 2 : Secondly, tables and figures should have a better layout.

RESPONSE 2: We have made some improvements to the layout of the tables and figures to make them more readable, especially by highlighting the p-values in the tables.

POINT 3 : As a final consideration, Conclusion section should be separate from the rest of the paper.

RESPONSE 3: We have separated the conclusion from the discussion section as a distinct part. We also revised the conclusion to better convey the results of the study.

Round 2

Reviewer 2 Report

The paper has been improved now, but the first lines of the Introduction section (e.g. from line 24 to line 30) still need to be re-written in order to give them a more scientific soundness. After this small modification, the paper can be considered for publication.

Author Response

POINT 1 : The paper has been improved now, but the first lines of the Introduction section (e.g. from line 24 to line 30) still need to be re-written in order to give them a more scientific soundness. After this small modification, the paper can be considered for publication.

RESPONSE 1: Thank you for reviewing our study. We have revised the introduction based on your advice. Some unnecessary or irrelevant parts have been removed, and the topic and purpose of the paper have been revised to better convey the message.
